# Canonical and Non-Canonical Localization of Tight Junction Proteins during Early Murine Cranial Development

**DOI:** 10.3390/ijms25031426

**Published:** 2024-01-24

**Authors:** Shermin Mak, Annette Hammes

**Affiliations:** 1Max-Delbrück-Center for Molecular Medicine in the Helmholtz Association (MDC), 13125 Berlin, Germany; shermin.mak@mdc-berlin.de; 2Institute for Biology, Free University of Berlin, 14159 Berlin, Germany

**Keywords:** mouse neural tube, forebrain, tight junctions, neurulation, ZO-1, claudins, occludin

## Abstract

This study investigates the intricate composition and spatial distribution of tight junction complex proteins during early mouse neurulation. The analyses focused on the cranial neural tube, which gives rise to all head structures. Neurulation brings about significant changes in the neuronal and non-neuronal ectoderm at a cellular and tissue level. During this process, precise coordination of both epithelial integrity and epithelial dynamics is essential for accurate tissue morphogenesis. Tight junctions are pivotal for epithelial integrity, yet their complex composition in this context remains poorly understood. Our examination of various tight junction proteins in the forebrain region of mouse embryos revealed distinct patterns in the neuronal and non-neuronal ectoderm, as well as mesoderm-derived mesenchymal cells. While claudin-4 exhibited exclusive expression in the non-neuronal ectoderm, we demonstrated a neuronal ectoderm specific localization for claudin-12 in the developing cranial neural tube. Claudin-5 was uniquely present in mesenchymal cells. Regarding the subcellular localization, canonical tight junction localization in the apical junctions was predominant for most tight junction complex proteins. ZO-1 (zona occludens protein-1), claudin-1, claudin-4, claudin-12, and occludin were detected at the apical junction. However, claudin-1 and occludin also appeared in basolateral domains. Intriguingly, claudin-3 displayed a non-canonical localization, overlapping with a nuclear lamina marker. These findings highlight the diverse tissue and subcellular distribution of tight junction proteins and emphasize the need for their precise regulation during the dynamic processes of forebrain development. The study can thereby contribute to a better understanding of the role of tight junction complex proteins in forebrain development.

## 1. Introduction

The development of the embryonic neural tube is a complex and highly coordinated process that is crucial for the establishment of the central nervous system. During primary neurulation, the neuroepithelium undergoes major changes on both a cellular and tissue level. Originating from the dorsal ectoderm during gastrulation, the neural plate begins as a simple neuroepithelial monolayer with the apical side exposed to the amniotic cavity and the basal side contacting the basal lamina and facing the underlying mesenchymal cells. With ongoing proliferation, the neural plate initially remains a monolayer but becomes pseudostratified. Subsequent patterning and morphogenetic events, involving cell segregation, bending of the neural plate, elevation, and convergence of the neural folds, lead to neural tube closure along the dorsal midline and finally to the formation of the brain and spinal cord [1,2,3,4,5,6]. Neurulation processes are not only dependent on the neuroectoderm but also involve surrounding tissues, including the non-neuronal ectoderm, mesoderm, and notochord, all of which cooperate in regulating neural tube closure (NTC). Disruptions in dynamic cell and tissue remodeling during neurulation can result in neural tube defects (NTDs), which are among the most common human birth defects [6,7]. The development of the vertebrate brain from the neuroepithelium is intimately connected with cranial neural crest development in time and space. Initially, morphogen-mediated patterning specifies a population of cells along the lateral edges of the neuroepithelium, the neural plate border (NPB). From the NPB, the neural crest is determined as an epithelial tissue in between the neuronal and non-neuronal ectoderm. During neurulation, neural crest cells come to lie within the neural folds bilaterally. Driven by epithelial-to-mesenchymal transition (EMT), neural crest cells delaminate from the border of the neuroepithelium to become migratory and populate their target tissues [8,9,10,11]. Defects in neural crest dynamics result in so-called neurocristopathies [12].

During early mouse neural tube development, tight junction proteins are key regulators for orchestrating cell polarity, tissue architecture, and signaling events, all essential for proper neural tube closure and neurogenesis.

Tight junctions are multiprotein junctional complexes and the resolution of electron microscopy revealed the structural complexity of these intercellular junctions [13,14]. Tight junctions are composed of integral membrane proteins, such as claudins, occludin, and junctional adhesion molecules (JAMs) as well as of cytoplasmic linker and adapter proteins, that serve as primary components of the apical junctional complex [15,16,17,18,19,20,21,22].

ZO-1 (zona occludens protein-1) is part of the zona occludens protein family and associated with the cytoplasmic side of tight junctions. ZO-1 serves as a scaffolding protein that interacts with the membrane integral tight junction proteins, such as occludin and claudins, crosslinking them at the junctional complex and anchoring them to the actin cytoskeleton [23,24,25].

Tight junctions play a pivotal role in establishing a selective barrier that restricts the diffusion of apical and basolateral membrane proteins, thereby governing membrane and cell polarity [18]. Additionally, tight junctions regulate the paracellular movement of ions and water across epithelial and endothelial cell layers, constituting a significant perm-selective barrier [26]. Tight junctions play essential functions in various physiological processes, including the establishment of tissue boundaries, the formation of blood–brain barriers, the regulation of nutrient absorption in the digestive tract, and the formation of a paracellular transport pathway for various ions to be reabsorbed by the kidney [27,28,29,30,31,32,33,34,35,36,37].

Dysfunction of tight junctions is implicated in various diseases, such as inflammatory bowel diseases, kidney diseases, blood–brain-barrier-associated diseases, neurodegenerative disorders, and multiple types of cancers [38,39,40,41,42,43,44]. In adults, disruptions in tight junction integrity can lead to increased permeability and contribute to the pathogenesis of these conditions [45].

While tight junctions are certainly crucial for maintaining tissue homeostasis in adult organs, they are also involved in regulating cell shape changes, patterning and morphogenesis during embryonic development [42,46,47]. The precise coordination of tight junction proteins is crucial for establishing the apical–basal polarity of neuroepithelial cells and maintaining neuroepithelial integrity, a prerequisite for proper neural tube development. As neural progenitor cells undergo proliferation, migration, and differentiation, tight junction proteins modulate intercellular adhesion and communication, ensuring the coordinated movement of cells within the developing neural tube. Thus, tight junctions are crucial for tissue remodeling, including junctional dynamics, epithelial-to-mesenchymal transition, and cell migration, that influence cell proliferation, polarization, cell fate determination, and differentiation [46,48,49,50,51]. Moreover, tight junctions can actively participate in signaling pathways as so-called signaling hubs [52,53]. Transmembrane proteins, associated with tight junctions, interact with cytoplasmic adaptors, regulatory proteins, and the cytoskeleton and thereby receive and transmit signals from the cell interior, further highlighting their multifaceted role during neurulation and neurogenesis. Therefore, it is not surprising that disruptions in tight junction function have been associated with neural tube defects, emphasizing the critical role tight junction proteins play in orchestrating proper neural tube closure [50,54,55].

So far, the analysis of tight junction proteins has been primarily performed in cell culture or the chicken model. Little is known about the complex tight junction composition in the developing early mouse neural tube. In this study, we try to shed light on the presence and localization of various tight junction proteins in several neurulation stages of mouse embryos, focusing on the cranial neural tube.

## 2. Results

### 2.1. Expression Profile of Tight-Junction-Related Proteins in Embryonic Mouse Heads

Bulk RNA sequencing of E 9.5 mouse embryonic heads revealed the expression of various proteins that are part of the tight junction complex (Figure 1). These transcriptome data were based on previous Illumina RNA library sequencing carried out in our laboratory [56]. mRNA-sequencing quantifications were obtained from exon-mapped and paired-end reads. The expression quantification process included read normalization, size factor estimation and differential expression analysis using DESeq2 v1.12.4, as described previously [57]. For this study, we filtered the transcriptome data from C57BL6/N wild-type samples to provide a comprehensive overview of the expression profile of all tight-junction-related genes at this early stage of cranial development. Among the classical claudin family members (claudins 1–10, 14, 15, 17, 19) [58], claudin-3, claudin-4, claudin-5, and claudin-6 exhibited higher expression levels compared to other family members, with claudin-6 demonstrating the highest expression levels. In the case of the non-classical claudins (claudins 11–13, 16, 18, 20–24) [58], claudin-12 displayed the highest relative expression levels. Claudin-25, a relatively recently identified family member [59], also showed robust expression levels. In addition to the claudin family members, we detected expression for other proteins of the tight junction complex with tight junction protein-1 (*Tjp1*, alias *Zo-1*) showing highest expression levels. Tight-junction-associated protein 1 (*Tjap1*) and junction adhesion molecule 3 (*Jam3*) were also abundantly expressed at the RNA level.

Tight junction proteins are typically located at the border between the apical and basolateral membranes. However, some tight junction proteins also localize to the basolateral membrane, intracellular cytoplasmic vesicles, or nucleus, rather than exclusively to the apical tight junction complex. These intracellular tight junction molecules likely play crucial roles in epithelial function beyond barrier and gating functions [60]. In addition to the analysis of the transcriptome profile, we performed immunohistochemistry for the tight junction complex proteins that exhibited the highest RNA expression levels to analyze their presence at the protein level as well as their tissue-specific and subcellular localization. For claudin-6, the available antibody did not work in our immunohistochemistry approach and, therefore, further data on claudin-6 are not currently available.

### 2.2. Localization of the Zona Occludens-1 (ZO-1) Protein during Mouse Cranial Neurulation

Zonula occludens-1 (ZO-1), also known as tight junction protein-1 (TJP1), is a 220-kD peripheral membrane protein that belongs to the family of zonula occludens proteins (ZO-1, ZO-2, and ZO-3), which are tight-junction-associated proteins [25,61,62].

The localization of ZO-1 was analyzed in mouse embryos at embryonic stage (E) 8.5, when the neural plate is not yet elevated (Figure 2A); and at E 9.0, when the neural folds are converging, shortly before neural tube closure (Figure 2B).

Signals detected for ZO-1 were strongest in the neuroectoderm, less strong in the non-neuronal ectoderm and mesenchymal cells facing the neuroepithelium, and absent in the neural crest cells (Figure 2A,B). Higher-magnification images confirmed ZO-1 localization at apical cell–cell junctions (Figure 2A′). En-face images on whole-mount neural fold samples at E 8.0 clearly showed the typical rosette formation of neuroepithelial cells (Figure 2C). Long and short junctions are present between cells of the rosettes. Between different junctions and along the same junctions, ZO-1 did not show a homogenous distribution but more intense signals at the multicellular junctions were observed (Figure 2C′). Cells with a larger cell surface area displayed less intense ZO-1 signals compared to apically constricted cells, likely reflecting the dynamic apical constriction process (Figure 2C′). A hallmark of neural tube closure is hinge point formation of the neural plate, which is dependent on apical constriction of the neuroepithelial cells. The apical surface constriction takes place in a pulsed ratchet-like fashion through asynchronous shrinkage of apical junctions [63,64,65]. Interestingly, in slightly later stages during neural tube formation when neural folds are elevating and converging, whole-mount immunofluorescence staining for ZO-1 could clearly demarcate the boundary between the neuronal and non-neuronal ectoderm (Figure 3B), highlighting the junctional dynamics during cranial neural tube formation.

### 2.3. Claudin-4 Is Specific to the Non-Neuronal Ectoderm in Cranial Mouse Neural Folds

Claudin-4 is an integral membrane protein, which belongs to the claudin family [66]. The protein is a component of the tight junction complex, and chromosomal microdeletions including the Claudin-3 and Claudin-4 genes are associated with Williams–Beuren syndrome, a neurodevelopmental disorder affecting multiple systems [67]. The most common symptoms of Williams–Beuren syndrome are congenital heart defects and unusual facial features, including a broad forehead, underdeveloped chin, and short nose.

In this study, claudin-4 localization was analyzed in E 8.5 mouse cranial neural folds. Claudin-4 was exclusively localized to the non-neuronal ectoderm overlapping with E-cadherin and clearly demarking the border to the neuronal ectoderm (Figure 3A). The en-face view of the cranial neural folds in whole-mount mouse embryos confirmed the non-neuronal ectoderm localization of claudin-4 (Figure 3B). In the co-staining analysis with ZO-1, the localization of claudin-4 exhibited a distinct punctate pattern along the cellular junctions, contrasting with the more evenly distributed pattern observed for ZO-1 (Figure 3B′).

### 2.4. Non-Junctional Localization of Claudin-3 in the Early Developing Neural Tube

In our bulk RNA sequencing screen, claudin-3 emerged as a prominently expressed gene. Subsequent co-staining with ZO-1 and claudin-3 in E 8.5 mouse neural folds showed no overlap between these two tight-junction complex proteins (Figure 4A). Instead, signals for claudin-3 overlapped with immunofluorescence signals for lamin B1, a nuclear membrane marker, in E 8.5 and E 9.5 mouse cranial neural folds (Figure 4B,C). Lamin B1 is one of the two B-type lamin proteins, which are components of the nuclear lamina, a fibrous layer on the nucleoplasmic side of the inner nuclear membrane. Compared to signals for lamin B1, claudin-3 showed a more punctate staining (Figure 4B′,C′). All cell types in the developing cranial neural folds, including neuroepithelial cells, non-neuronal ectoderm cells and mesenchymal cells, showed claudin-3 localized to the nuclear membrane. Two different antibodies, widely used in the field, were used and specificity was validated on sections from claudin-3 null mutants (Figure 5A). In addition, we detected classical apical junction localization for claudin-3 in tissue sections from later-stage mouse embryos (Figure 5B,C). Immunohistochemistry at E 13.5 and E 18.5 detected claudin-3 localization at the apical junctions in various tissues, including in the colon and salivary glands (Figure 5B′,C′,C″). Of note, during these embryonic stages, claudin-3 signals were no longer detectable in the developing mouse forebrain region (Figure 5B,C).

### 2.5. Claudin-1 and Occludin Are Localized to the Neuroepithelium and to the Non-Neuronal Ectoderm

One of the most abundant claudins is claudin-1. Mice deficient for claudin-1 die shortly after birth [69]. In E 8.5 mouse cranial neural folds, we observed the most intense claudin-1 signals in the neuroepithelium, weaker signals in the non-neuronal ectoderm, and minimal signals in mesenchymal cells (Figure 6A). Occludin, an integral membrane protein, forms the tight junctions in conjunction with claudin family members [70]. Similar to claudin-1, we detected occludin signals mainly in the neuroepithelium and non-neuronal ectoderm of the cranial neural folds at E 8.5 (Figure 6B). Mesenchymal cells showed very weak occludin staining (Figure 6B). Compared to claudin-1, occludin showed more prominent apical staining and a weaker signal in the basolateral domain (Figure 6A,B). En-face views onto whole-mount neural folds (only one hemisphere shown here) demonstrated slightly different localization for claudin-1 and occludin along the junctions. While claudin-1 signals showed a more meshwork-like pattern with higher intensity at the multicellular junctions, occludin displayed a more punctate pattern along the junctions (Figure 6C,C′).

### 2.6. Localization of Claudin-12 Exclusively in Neuronal Tissue and Localization of Claudin-5 Only in SOX10-Negative Mesenchymal Cells

Claudin-5 and claudin-12 were among those claudin family members showing higher expression levels in the developing cranial neural tube (Figure 1).

For claudin-5, we detected only very faint signals in the neuroepithelium but robust expression in the SOX10-negative mesenchymal cells (Figure 7A) in E 8.5 mouse neural folds. These claudin-5 positive mesenchymal cells are mesoderm-derived cells underlying the neuroepithelium. SOX10 immunohistochemistry was applied here to label neural crest cells. SOX10 (SRY-related HMG-box) is a transcription factor and is specifically expressed in migrating neural crest cells that have delaminated from the dorsal neural tube [71].

For claudin-12, we demonstrated its presence exclusively in the neuroepithelium (Figure 7B). Claudin-12 showed, besides ZO-1 (Figure 2), the clearest apical junction localization (Figure 7B,B′). Interestingly, all claudins have a conserved PDZ binding motif at their C-termini, except for claudin-12 [50]. The PDZ binding motif interacts with the PDZ domain of ZO-1, ZO-2, ZO-3, and other tight junction cytoplasmic proteins. While the role of claudin-12 has been studied in the context of the blood–brain barrier function in the adult murine brain [72], little is known about claudin-12, as one of the non-canonical claudins, in the developing neural tube.

## 3. Discussion

The formation of the head, including brain and face, encompasses one of the most complex developmental endeavors. During neurulation and in particular during formation of the cranial neural tube, which shapes the head, the neuroepithelium has to undergo major changes. The neural tube starts as a flat neuroepithelial sheet, which then proliferates and forms a pseudostratified neuroepithelium (neuroectoderm). The neuroectoderm is laterally connected to the squamous non-neuronal ectoderm, the later epidermis [3]. The initially flat neural plate bends at the ventral midline and forms the opposing neural folds. These neural folds have to elevate and form hinge points to finally meet and fuse at the dorsal midline and form the neural tube. Simultaneously, the development of the optic vesicles requires evagination of the neural plate. Moreover, epithelial-to-mesenchymal transition at the neural plate border is required for neural crest development and is therefore indispensable for head and face formation. During early neurulation, the neuronal and non-neuronal ectoderms separate from each other and express distinct markers, such as N-cadherin and E-cadherin, respectively. At the end of neurulation, the non-neuronal ectoderm, laterally located to both neural folds, also fuses at the dorsal midline covering the neural tube. Throughout neurulation, interactions between the neuronal and non-neuronal ectoderm are important for proper neural crest induction and neural tube formation [73]. These intricate complex cranial morphogenetic processes must be integrated and coordinated simultaneously during neurulation. Neuroepithelial cells exhibit apical–basal polarity as well as planar polarity. Several studies have shown that proteins of tight junction complex are essential for neuroepithelial integrity, including apical–basal polarity, and ultimately for neural tube closure [50,55,74,75,76,77,78]. However, limited information is available regarding the tight junction complex proteins in the anterior (cranial) neural tube in the mouse. In this study, we show distinct expression patterns for several tight junction proteins in the developing cranial mouse neural folds. Notably, we identified overlapping yet clearly distinct compositions of tight junctions between non-neuronal and neuronal ectoderm tissues.

### 3.1. Expression Profile of Genes Encoding Tight Junction Complex Proteins

On the RNA level, we identified multiple genes encoding proteins of the tight junction complex in the embryonic cranial mouse neural tube. Among the classical claudin family members [58], we observed robust RNA expression levels for claudin-3, claudin-4, claudin-5, claudin-6 and claudin-7. For most of these claudins, we conducted protein presence and localization tests in the neural tube, and the results are discussed below. Interestingly, claudin-6 showed the highest expression levels. Unfortunately, the immunohistochemistry approach using the available claudin-6 antibody requires optimization, and results on the tissue-specific and subcellular localization of claudin-6 in the mouse cranial neural tube are not currently available. Claudin-6 recently gained attention as a novel target for CAR-T cell cancer therapy [79]. Claudin-6 is found in several epithelial tissues in neonatal mice [80] and has been demonstrated to be essential for blastocyst formation in mouse embryos [81] and for the induction of epithelial morphogenesis [82]. Another study reported the presence of claudin-6 in cells that express SOX2 in the developing mouse lung [83]. SOX2 (sex-determining region Y (SRY)-box 2) is also a prominent marker in the early mouse neuroepithelium and is crucial for neural stem cell function [84]. Therefore, it is plausible to hypothesize that claudin-6 is present in neuroepithelial cells of the developing cranial mouse neural tube at E 8.5. Of note, a previous study has shown that claudin-6 is a highly specific cell surface marker of human pluripotent stem cells and is downregulated during differentiation into neural lineages [85]. We therefore suggest that claudin-6 has a function in neuroepithelial cells before the onset of neurogenesis.

Several non-classical claudins were also identified on the RNA level in the cranial neural tube, with claudin-12 exhibiting the highest expression levels. Interestingly, claudin-11 showed relatively low expression levels. The observed low expression levels for claudin-11 at E 9.5, before the differentiation of radial glia cells to oligodendrocyte precursors, is in line with previously published data describing claudin-11 mRNA in developing meninges at E 15.5 and in oligodendrocytes of the brainstem at E 18.5 [86]. Postnatally, claudin-11 is expressed predominantly by oligodendrocytes in the brain [86,87,88]. In addition to the claudin family members, we detected expression for other members of the tight junction complex, with tight-junction-associated protein 1 (*Tjap1*) and junction adhesion molecule 3 (*Jam3*) abundantly expressed at the RNA level. In the adult brain, TJAP-1 and JAM3 (alias JAM-C) are important for blood–brain barrier integrity and were identified as potential therapeutic targets of the protective microRNA-132/212 after ischemic stroke [89]. Moreover, mice deficient for JAM-3 develop a severe hydrocephalus, suggesting a role of this tight junction complex protein in brain development and cerebrospinal fluid homeostasis [90,91]. However, the roles of TJAP-1 and JAM3 during early forebrain development remain to be investigated.

### 3.2. Tight-Junction-Associated Proteins in the Neural Tube

Zonula occludens-1 (ZO-1, alias TJP1) is a tight-junction-associated protein. ZO-1 is widely expressed in embryonic tissue, and mice deficient for ZO-1 are embryonic lethal [92]. We found ZO-1 in non-neuronal and neuronal epithelia of the neural tube with stronger signals in the neuroepithelium similar to claudin-1. However, neural plate border neuroepithelial cells showed fewer ZO-1 signals, suggesting significant changes in tight junction formation in this region, which serves as the niche for neural crest stem cells during neurulation and concurrent neural crest cell delamination.

### 3.3. Occludin in the Cranial Neural Tube

Occludin, along with members of the claudin family, is a major component of the tight junction. Of note, occludin (*Ocln*) exhibited relatively low RNA expression levels in mouse embryonic heads at E 9.5 after neural tube closure (Figure 1). However, at earlier embryonic stages (E 8.5) preceding neural tube closure, we detected robust signals for occludin using immunohistochemistry in the cranial neural tube (Figure 6). These findings are in line with data from the Huttner laboratory. Aaku-Saraste and colleagues reported that immunoreactivity for occludin in the chicken neural tube was detected at the apical junction of neuroepithelial cells during neurulation but disappeared after neural tube closure [74]. This is reminiscent of findings for claudin-1 in the chicken [93], where claudin-1 was also detected in the neural tube before but not after closure. The data suggest a synergistic function, likely in paracellular barrier formation, of claudin-1 and occludin during neurulation processes before neural tube closure when the neuroepithelium is still exposed to the amniotic cavity. The dynamic expression of tight junction proteins, particularly occludin, during neural tube closure processes underscores the significance of molecular remodeling at the intercellular apical junctions. While it seems to be widely accepted that occludin expression in the embryonic cortex is limited to neuroepithelial junctions prior to the neuroepithelial-to-radial glia cell transition at the onset of neurogenesis, recently published data suggest that occludin is still expressed in an isoform-specific manner at the junctions and centrosomes, respectively, during later stages of CNS development, where it regulates progenitor self-renewal and survival in the developing cortex [94].

### 3.4. Claudins in the Cranial Neural Tube

The claudin family of tetraspan transmembrane proteins is, together with occludin, essential for tight junction formation in epithelial cells. Claudins and occludin play a role in apical–basal cell polarity and cell adhesion, and they are essential to link the tight junction complex to the actin cytoskeleton. Bridging the cellular junction and epithelial circumferential actin belt is a prerequisite for epithelial cell shape changes that facilitate complex neural fold morphogenesis.

#### 3.4.1. Non-Neuronal Ectoderm-Specific Claudin

Claudin-4, one of the classical claudins, is a marker for epithelial differentiation [95]. Claudin-4 is involved in tight junction formation in epithelial cells, including those in the intestines and lungs, and is associated with many epithelial malignancies [66,96,97]. In our study, we showed that claudin-4 was exclusively localized to the non-neuronal ectoderm, overlapping with E-cadherin. Non-neuronal ectoderm localization in the cranial mouse neural folds during neurulation could reflect an important role of this tight junction protein in defining non-neuronal versus neuronal ectoderm fate and, consequently, in facilitating proper interaction between these tissues during head and face development. The accurate segregation between non-neuronal and neuronal ectoderm at the neural plate border is essential for neural crest specification. Of note, claudin-4 has been associated with Williams–Beuren syndrome (WBS), a neurodevelopmental disorder, with unusual facial features including a broad forehead, underdeveloped chin, and short nose. WBS is caused by a heterozygous microdeletion in chromosome 7, and genes identified within the common WBS deletion include claudin-3 and claudin-4 [67,98]. However, a clear correlation between the potential claudin-3, -4 haploinsufficiency and a specific WBS pathomechanism has not been established so far.

A study by Werth et al. provided evidence that the transcription factor grainyhead-like 2 (GRLH2), expressed in the non-neuronal ectoderm but not in the neuronal ectoderm, regulates the expression of claudin-4 [99]. The authors showed that *Grlh2* null mutant mice display reduced claudin-4 expression levels in the non-neuronal ectoderm. Interestingly, these mice suffer from neural tube closure defects, underscoring the importance of mutual inductive signals between neuronal and non-neuronal ectoderm for neural tube closure processes.

#### 3.4.2. Non-Canonical Localization of Claudins in the Neural Tube

Claudin-3 is another classical claudin family member. In our study, we demonstrated claudin-3 localization to the nuclear membrane in the neuroepithelium, the non-neuronal ectoderm, and mesenchymal cells during cranial neural tube development at E 8.5 and E 9.5. Claudin-3 was localized to the apical tight junction in epithelia of several tissues, including the colon and salivary gland, in later-stage mouse embryos, where it serves the classical barrier function. However, its function during mouse neurulation, where claudin-3 showed nuclear membrane localization, remains unclear. In a previously published study by Baumholtz et al., the authors showed that simultaneous removal of claudin-3,-4 and -8 from tight junctions by applying the C-terminal domain of C. perfringens enterotoxin (C-CPE) caused folate-resistant open neural tube defects in chicken. C-CPE-treated mouse embryos showed milder effects on neural tube closure [50]. A very recent study from the same laboratory showed that claudin-3 deficiency in the non-neuronal ectoderm causes spinal neural tube defects [78]. The studies by Baumholtz et al. and Legere et al. were mainly based on the chicken model, which seems to display a slightly different expression pattern for claudins, in particular claudin-3 and claudin-4 during neurulation, compared to the mouse model [99]. It is also possible that the expression pattern of claudins differs between the caudal and cranial neural tube. Nuclear staining for claudin-3 and also for claudin-1 has been reported before, mainly in the context of tumors. Claudin-3 was detected in the nucleus of breast cancer cell lines [100] and colorectal adenocarcinomas [101]. Nuclear localization of claudin-1 was detected in primary colon carcinomas and metastasis [102]. These studies not only propose nuclear claudin localization as a potential histopathological biomarker but also imply a causal link between claudin expression/localization and cellular transformation. In this context, another study has provided interesting data, showing that transforming growth factor β (TGFβ) signaling can lead to redistribution of claudin-3 from tight junctions into nuclei [103]. Overall, accumulating evidence suggests that claudins, as well as ZO-1 and ZO-2, extend beyond their traditional roles in cell–cell adhesion, fence and barrier formation. Several studies indicate their involvement in transcriptional regulation [52,60,104,105,106]. Interestingly, several claudins contain a putative nuclear localization sequence [60]. For nuclear claudin-1, a direct role in E-cadherin transcriptional regulation, affecting the ß-catenin/TCF/LEF signaling pathway, has been demonstrated [102]. Whether nuclear claudins can directly regulate transcription by binding to the DNA or through indirect pathways remains unclear. Of note, we detected claudin-3 in the nuclear lamina. The nuclear lamina plays a crucial role in genome organization and chromatin structure. Additionally, it regulates gene expression by sequestering transcription factors at the nuclear envelope, limiting their availability in the nucleoplasm. Furthermore, the lamina serves as a platform for assembling protein complexes involved in signal transduction pathways [107]. Therefore, it is possible that claudin-3 in the nuclear lamina indirectly affects signaling and/or transcription by interacting with proteins regulating these processes.

#### 3.4.3. Mesoderm-Specific Claudins in the Cranial Neural Tube

Claudin-5 is the most enriched tight junction protein at the blood–brain barrier, and its dysfunction has been implicated in Alzheimer’s disease, multiple sclerosis and psychiatric disorders, including depression and schizophrenia [108]. Homozygous mutation of the claudin-5 gene in mice results in size-selective loosening of the blood–brain barrier [109]. Claudin-5 null mutant mouse neonates gradually cease movement and die shortly after birth. In this study, we detected only very faint signals for claudin-5 in the neuroepithelium but robust expression in the SOX10-negative mesoderm-derived mesenchymal cells underlying the neuroepithelium (Figure 7A) in E 8.5 mouse neural folds. This suggests that claudin-5 plays a minor role during neurulation within the neuroectoderm tissue and plays a more prominent role during later developmental stages.

#### 3.4.4. Classical Tight Junction Localization of Claudins and Occludin in the Non-Neuronal and Neuronal Ectoderm

Claudin-1, the first identified member of the claudin family, is strongly expressed in the kidney, intestine, spleen, liver, testis and brain [17,44,110]. Claudin-1 is required for maintaining normal water homeostasis and preventing excessive water loss through the skin. Mice homozygous for a mutation in claudin-1 have wrinkled skin and do not survive beyond one day after birth [69]. Additionally, several studies provide evidence for claudin-1 being one of the most deregulated claudins in human cancer. Claudin-1 can function as a tumor promoter or suppressor depending on the type of cancer [44].

During cranial neural tube development, we detected robust levels of claudin-1 in the neuroepithelium, weaker signals in the non-neuronal ectoderm and very weak signals in mesenchymal cells (Figure 6A), suggesting a more important role for claudin-1 in neuronal tissue during neurulation. This is in line with findings in the chicken, where claudin-1 was reported to be highly expressed in the neuroepithelium during open neural tube stages but decreased after neural tube closure [75,93]. The study by Fishwick et al. also showed that depletion or overexpression of claudin-1 in the neuroepithelium increases or reduces neural crest cell emigration, respectively, suggesting that the role of claudin-1 is not restricted to facilitate neuronal fates but is also involved in neural crest specification at the neural plate border. In our studies, claudin-1 showed localization at the apical junctions but also in the basolateral domain. Therefore, it remains unclear whether this claudin family member has functions beyond cell–cell adhesion and barrier formation. To date, it is unclear whether basolateral membrane claudins might serve as a reservoir of claudin molecules available for recycling to the apical tight junction complex when required or whether they serve a different function.

#### 3.4.5. Neuronal Ectoderm-Specific Claudins in the Cranial Neural Tube

Claudin-12, as a non-classical member of the claudin family lacking the PDZ binding motif that interacts with the ZO-1, ZO-2, and ZO-3 proteins, showed neuronal ectoderm-specific and classical apical junction localization in the developing cranial neural tube, resembling the ZO-1 pattern. Claudin-12-deficient mice survive and exhibit only mild and mostly sex-specific alterations associated with the central nervous system, energy metabolism and cardiovascular functions [72]. The lack of obvious neural tube defects in claudin-12 null mutants suggests redundant functions of claudin family members during neural tube development. It is possible that claudin-1 and/or claudin-6 can compensate for the loss of claudin-12 during neural tube development. In line with this assumption that claudins have functionally redundant roles in the neuroepithelium during neural tube closure, neural tube defects have not been reported in any of the single claudin null mutant mouse lines [69,111,112,113,114]. However, simultaneous removal of claudin family members in the chicken model by C-CPE treatment caused neural tube closure defects [50].

In summary, our study revealed a distinct tissue-specific and subcellular composition of various tight junction complex proteins in the early mouse forebrain. Claudin-4 showed exclusive expression in the non-neuronal ectoderm, while claudin-12 exhibited neuronal ectoderm-specific localization in the developing forebrain. Claudin-5 was uniquely present in mesenchymal cells. Claudin-1, occludin, and ZO-1 were present in both non-neuronal and neuronal tissue in the cranial neural tube. Regarding subcellular distribution, claudin-1 and occludin were detected in the apical junction as well as in basolateral domains, while claudin-12 and ZO-1 were predominantly localized to the apical junction domain. The apical tight junction complex is known to be highly dynamic and to constantly undergo remodeling [115]. Therefore, the basolateral claudins might serve as a pool of claudin molecules available for recycling to the apical tight junction complex when required [60], although the role of basolateral localization of tight junction proteins remains poorly understood. The differences in the subcellular pattern between different tight junction complex proteins might reflect their distinct dynamic behaviors. The results of this study contribute to a better understanding of the role of tight junction complex proteins in forebrain development. However, to decipher distinct and redundant functions of tight junction complex proteins during mouse neurulation, complex gene targeting experiments are needed. Moreover, employing fluorophore tagging on tight junction complex proteins for real-time high/super-resolution imaging in neural fold explant cultures could elucidate the distinct dynamic behaviors of these proteins within the developing neuroepithelium.

## 4. Materials and Methods

All experiments were performed in compliance with the relevant ethical regulations at the Max Delbrück Center for Molecular Medicine (MDC). All animal procedures were performed according to institutional animal welfare guidelines and regulations approved by the local authorities.

### 4.1. Animal Work and Embryo Collection

Animals were kept under pathogen-free conditions in a 12 h light/dark cycle (6:00 a.m.–6:00 p.m.). C57BL/6N mice from eight weeks old onwards were used for timed mating to collect embryos at the desired developmental stages. The gestational ages of pregnant females were determined based on the observation of vaginal plugs. The day when a vaginal plug was noted was considered as E 0.5. Embryos were dissected from the uteri of pregnant females in phosphate-buffered saline (PBS) on ice and then further staged by somite counting during dissection. Embryos were cut at the level below the heart during dissection to keep only the tissue of interest, cranial neural tube tissues.

### 4.2. Embryo Fixation and Cryosectioning

Dissected embryos were fixed with 4% paraformaldehyde (PFA) at room temperature for 1 h and then washed two times for 10 min each with PBS. Embryos were stored in PBS until use. For cryosectioning, samples were incubated gradually at room temperature for 30 min each in 15% and 30% sucrose in PBS. Tissues were then embedded within Optimal Cutting Temperature (O.C.T) (Tissue-Tek^®^, Sakura Finetek, Torrance, CA, USA) in plastic embedding molds of 10 mm × 10 mm × 5 mm in size and then frozen in a 100% methanol bath on dry ice. Fresh-frozen O.C.T blocks were mounted onto a cutting block containing a small amount of O.C.T in the cryostat and then sectioned coronally at 10 µm. Sections were stored at −20 °C until use.

### 4.3. RNA Library Generation and Sequencing

RNA expression data for tight junction proteins (Figure 1) were based on previously published RNA bulk RNA sequencing data [56]. In brief, for the generation of an RNA library, embryonic heads of E9.5 embryos were dissected, snap frozen and stored at −80 °C until RNA isolation. RNA was extracted using a RNeasy Plus Micro kit (Qiagen, 40724 Hilden, Germany, Cat. #74034), checked on Bioanalyzer (Agilent Technologies, Waldbronn, Germany, RNA 6000 Nano kit, Cat. #5067-1511), and samples with RIN > 9 were used to prepare the cDNA library. Library preparation for mRNA sequencing was performed according to the Illumina SR TruSeq Stranded mRNA protocol (Cluster Kit V3, Illumina GmbH, Munich, Germany, Cat. #20020594) on embryonic head RNA samples from mice with 24 somites. Sequencing was performed on the Illumina HiSeq2000 system with HCS 2.2.38 software. Sequencing read alignment and counting of the number of reads mapped to a gene were performed as described before [56].

### 4.4. Antibodies

All primary and secondary antibodies used in this article are listed in Table 1.

### 4.5. Immunofluorescence

#### 4.5.1. Sections

Tissue sections were dried at room temperature for 30 min before proceeding with the standard immunofluorescence staining protocol. In brief, tissues were washed four times for 7 min each in PBS with 0.1% Triton X-100 (PBST) at room temperature. Blocking was then performed in 10% donkey serum in 0.1% PBST for 1 h at room temperature. Tissues were incubated with primary antibodies (Table 1) in 0.1% PBST solution containing 1% donkey serum overnight at 4 °C. Following five washes of 7 min each with 0.1% PBST at room temperature, tissues were incubated with secondary antibodies (Table 1) diluted in 0.1% PBST for 1 h at room temperature. All samples were also counterstained with DAPI. Secondary antibodies were washed away with 0.1% PBST in another round of five washes for 7 min each at room temperature before tissues were mounted with DAKO fluorescent mounting medium (Agilent, S302380-2).

#### 4.5.2. Whole-Mount

Embryos were first permeabilized with 0.1% PBST for 15 min at room temperature before blocking with 1% donkey serum and 2% bovine serum albumin (BSA) in 0.1% PBST overnight at 4 °C. Incubation with primary antibodies was performed for 48 h at 4 °C in blocking solution. Following primary antibody incubation, embryos were washed five times for 1 h each at room temperature before transferring into secondary antibody solution diluted in blocking solution for overnight incubation at 4 °C. All samples were also counterstained with DAPI. Embryos were washed five times for 1 h each at room temperature to remove any unbound secondary antibodies. Embryos were mounted between two high-precision cover slips separated using Secure-Seal^TM^ Spacer (Invitrogen^TM^, S24735) in SlowFade^TM^ Diamond Antifade Mountant (Invitrogen^TM^, S36972).

### 4.6. Microscopy and Image Analysis

All images were acquired using a Leica TCS SP8 confocal microscope equipped with Leica LASX 3.0.0.15697 software. For section staining, images were processed using ImageJ 2.14.0 (Fiji, NIH, Bethesda, Maryland, USA, https://imagej.net/ij/). For whole-mount immunofluorescence staining, 3D images were imported into Huygens Professional 23.10 software (Scientific Volume Imaging B.V. Marathon 9E, 1213 PE Hilversum, Netherlands) to improve the signal-to-noise ratio and axial/spatial resolution by applying deconvolution with the CMLE algorithm and background correction. Deconvolved images were then visualized and processed using IMARIS 10.1 software (Bitplane, Zürich, Switzerland).

## Figures and Tables

**Figure 1 ijms-25-01426-f001:**
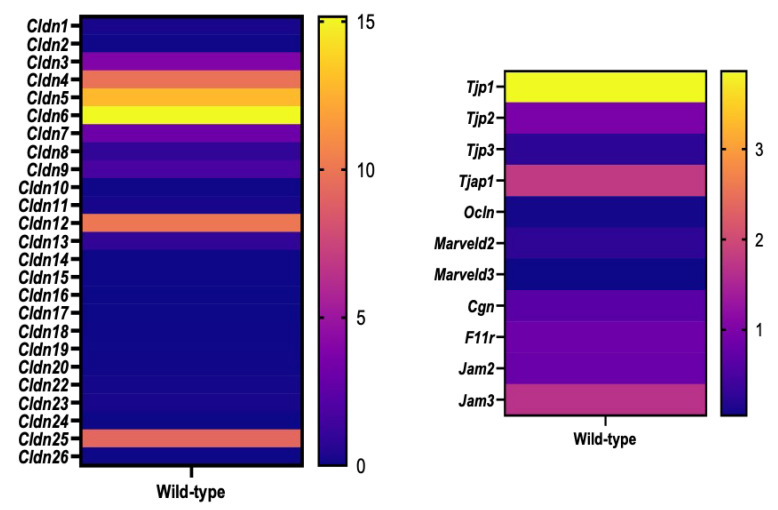
Expression profile of tight-junction-related proteins in mouse embryonic heads at E 9.5. Heat map of expression profiles for genes encoding claudin family members (**left panel**) and other tight junction complex proteins (**right panel**) based on bulk RNA sequencing of E 9.5 C57BL/6N mouse embryonic head tissue. Dark blue represents lowest relative expression levels; yellow represents highest relative expression levels. A total of 4 heads from somite matched embryos (24 somite stage) were individually submitted to Illumina sequencing. Abbreviations: *Cldn*: claudin (claudin 1–20 and 22–26); *Tjp*: tight junction protein (*Tjp1*, *Tjp2*, *Tjp3* alias zona occludens proteins ZO-1, ZO2, ZO-3); *Tjap1*: tight-junction-associated protein 1; *Ocln*: occludin; *Marveld*: MARVEL (membrane-associating) domain containing; *Marveld2* (alias tricellulin) and *Marveld3*; *Cgn*: cingulin; *F11r*: F11 receptor (alias Jam1); *Jam*: junction adhesion molecule (*Jam2*, *Jam3*).

**Figure 2 ijms-25-01426-f002:**
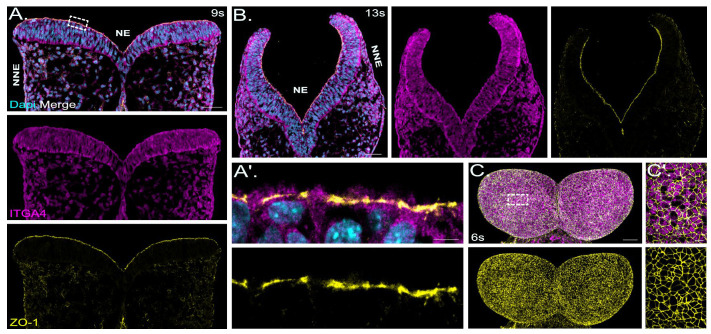
ZO-1 shows the strongest expression on the apical surface of the mouse neuroepithelium. (**A**): Confocal images show a representative coronal section of mouse cranial neural folds with immunofluorescence co-staining detecting ZO-1 (yellow) and integrin alpha 4 (ITGA4, magenta) at embryonic stage E 8.5 (9 somites, 9s). The overlay image of all channels (merge) is shown at the top of panel (**A**), followed below by the individual channel images, showing the signals for ITGA4 (magenta) and ZO-1 (yellow). Nuclei are stained with DAPI (cyan). Scale bar: 50 µm. Immunofluorescence staining for ITGA4 was performed to label the basolateral cellular domains. ZO-1 showed the strongest signals in the neuroectoderm (NE) and less strong signals in the non-neuronal ectoderm (NNE) and the mesenchymal cells, underlying the neuronal ectoderm. (**A′**): Higher magnification of boxed area in (**A**). The overlay image of all channels is shown at the top of panel (**A′**), followed below by the individual channel image, showing the signals for ZO-1 (yellow). Scale bar: 5 µm. ZO-1 was localized to the apical cell–cell junctions. (**B**): Confocal images show representative coronal section of mouse cranial neural folds with immunofluorescence co-staining detecting ZO-1 (yellow) and ITGA4 (magenta) at embryonic stage E 9.0 (13 somites, 13s). The overlay image of all channels is shown at the left of panel (**B**), followed to the right by the individual channel images, showing the signals for ITGA4 (magenta) and ZO-1 (yellow). Nuclei are stained with DAPI (cyan). Scale bar: 50 µm. ZO-1 showed the strongest signals in the neuroectoderm (NE) and less strong signals in the non-neuronal ectoderm (NNE) and mesenchymal cells. (**C**): E 8.0 whole-mount mouse embryos (6 somites, 6s) were immunofluorescence co-labelled for ZO-1 (yellow) and ITGA4 (magenta). The overlay image of both channels is shown at the top of panel (**C**), followed below by the individual channel image showing the signals for ZO-1 (yellow). The frontal view on the whole-mount forebrains, imaged using confocal microscopy, is shown. Scale bar: 50 µm. (**C′**): Magnification of boxed area indicated in (**C**) with merged-channel image at the top, showing ITGA4 (magenta) and ZO-1 (yellow) signals and ZO-1 single-channel image below. Scale bar: 10 µm.

**Figure 3 ijms-25-01426-f003:**
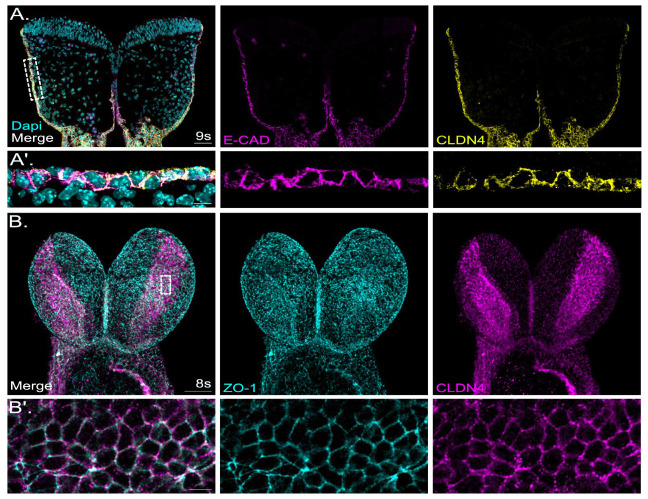
Claudin-4 is specifically localized to the tight junctions of non-neural ectoderm in the developing mouse neural tube. (**A**): Images show a representative coronal section of mouse cranial neural folds with immunofluorescence co-staining detecting E-cadherin, a non-neuronal ectoderm marker (E-cadherin, E-CAD, magenta), and claudin-4 (CLDN4, yellow) at embryonic stage E 8.5 (9 somites, 9s). Nuclei are stained with DAPI (cyan). The overlay image of all channels (merge) is shown at the left of panel (**A**), followed to the right by the individual channel images, showing the signals for E-CAD (magenta) and CLDN4 (yellow). Scale bar: 50 µm. Claudin-4 was specifically localized to the non-neuronal ectoderm (NNE) overlapping with E-CAD. No signal was observed in the neuroepithelium. (**A′**): Higher-magnification images of boxed area in (**A**). Scale bar: 10 µm. Claudin-4 showed overlapping signals with E-cadherin. (**B**): E 8.5 whole-mount mouse embryos (8 somites, 8s) were immunofluorescence co-labelled for ZO-1 (cyan) and claudin-4 (CLDN4, magenta). The overlay image of both channels (merge) is shown at the left of panel (**B**), followed to the right by the individual channel image, showing the signals for ZO-1 (cyan) and CLDN4 (magenta). The frontal view on the whole-mount forebrains, imaged using confocal microscopy, is shown. Scale bar: 50 µm. (**B′**): Higher-magnification images of boxed area indicated in (**B**). Scale bar: 10 µm. Claudin-4 was specifically localized to the non-neuronal ectoderm (NNE). No signal was observed in the neuroepithelium.

**Figure 4 ijms-25-01426-f004:**
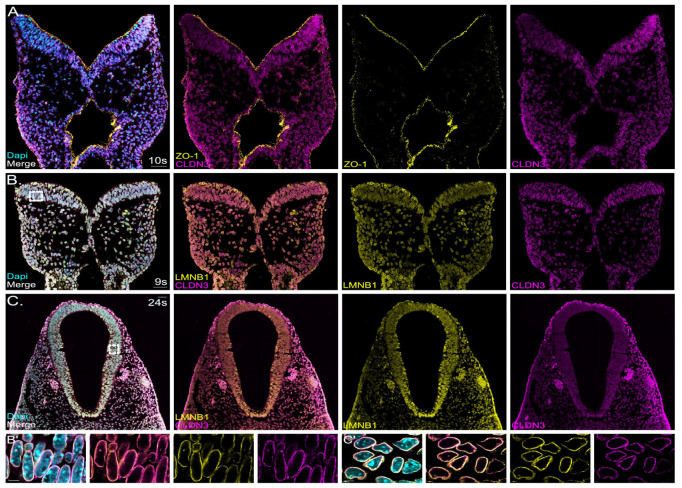
Claudin-3 shows non-junctional localization in the early embryonic stages. (**A**): Confocal images show a representative coronal section of mouse cranial neural folds with immunofluorescence co-staining detecting ZO-1 (yellow) and claudin-3 (CLDN3, magenta) at embryonic stage E 8.5 (10 somites, 10s). The overlay image of all channels (merge) is shown at the left of panel (**A**), followed to the right by the overlay image for ZO-1 and CLDN3 and the individual channel images, showing the signals for ZO-1 (yellow) and CLDN3 (magenta). Nuclei are stained with DAPI (cyan). Scale bar: 50 µm. Claudin-3 showed non-junctional localization in the neuroectoderm, in non-neuronal ectoderm and in mesenchymal cells. (**B**): Confocal images show representative coronal section of mouse cranial neural folds with immunofluorescence staining detecting lamin b1 (LMNB1, yellow) and claudin-3 (CLDN3, magenta) at embryonic stage E 8.5 (9 somites, 9s). The overlay image of all channels (merge) is shown at the left of panel (**B**), followed to the right by the overlay image for LMNB1 and CLDN3 and the individual channel images, showing the signals for LMNB1 (yellow) and CLDN3 (magenta). Nuclei are stained with DAPI (cyan). Scale bar: 50 µm. (**B′**): Higher-magnification images of boxed area in (**B**). Scale bar: 5 µm. Claudin-3 showed overlapping signals with lamin B1 around the nuclei. (**C**): Confocal images show representative coronal section of mouse cranial neural folds with immunofluorescence staining detecting lamin b1 (LMNB1, yellow) and claudin-3 (CLDN3, magenta) at embryonic stage E 9.5 after neural tube closure (24 somites, 24s). The overlay image of all channels (merge) is shown at the left of panel (**B**), followed to the right by the overlay image for LMNB1 and CLDN3 and the individual channel images, showing the signals for LMNB1 (yellow) and CLDN3 (magenta). Nuclei are stained with DAPI (cyan). Scale bar: 50 µm. (**C′**): Higher magnification of boxed area in (**C**). Scale bar: 5 µm. Claudin-3 showed overlapping signals with lamin B1 around the nuclei.

**Figure 5 ijms-25-01426-f005:**
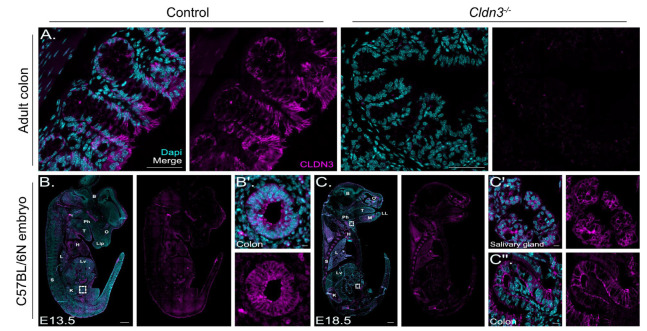
Apical junction localization of claudin-3 in mouse tissue. (**A**): Confocal images show representative section of adult mouse colon tissue from wild-type controls and claudin-3^-/-^ mice [68], respectively. Immunohistochemistry detecting claudin-3 (CLDN3, magenta) showed apical localization in colon epithelial cells of control. Nuclei are stained with DAPI (cyan). Merged- and single-channel images are shown at the left and right, respectively, for control and claudin-3^-/-^ (*Cldn3^-/-^*) samples. Scale bar: 50 µm. No claudin-3 signals were detected in tissue from claudin-3^-/-^ mice. (**B**): Images show a representative sagittal section of a mouse embryo at E 13.5. Claudin-3 signals (magenta) were detected in several tissues including in the colon (boxed area). Nuclei are stained with DAPI (cyan). Merged- and CLDN3 single-channel images are shown at the left and right, respectively. Scale bar: 500 µm. Further tissues, positive for claudin-3, included heart (H), lung (L), and kidney (K). (**B′**): Higher magnification of the boxed region in (**B**), showing CLDN3 signals in the colon. Scale bar: 10 µm. Abbreviations: B: brain, Ph: pharynx, T: tongue, Lip: upper lip, H: heart; L: lung, Lv: liver, S: spine, K: kidney. (**C**): Images show a representative sagittal section of a mouse embryo at E 18.5. Claudin-3 signals (magenta) were detected in several tissues including in the salivary gland (cranial boxed area, see (**C′**)) and colon (caudal boxed area, see (**C″**)). Nuclei are stained with DAPI (cyan). Merged- and single-channel images are shown at the left and right, respectively. Scale bar: 1000 µm. Further tissues, positive for claudin-3, included the olfactory system (O), lung (L), kidney (K). **C′**: Higher magnification of the boxed region in the cranial part of the embryo in (**C**). Scale bar: 10 µm. (**C″**): Higher magnification of the boxed region in the caudal part of the embryo in (**C**). Scale bar: 10 µm. Abbreviations: B: brain, O: olfactory system; T: tongue, Ph: pharynx, M: mandible, LL: lower lip, H: heart; L: lung, S: spine, Lv: liver, K: kidney.

**Figure 6 ijms-25-01426-f006:**
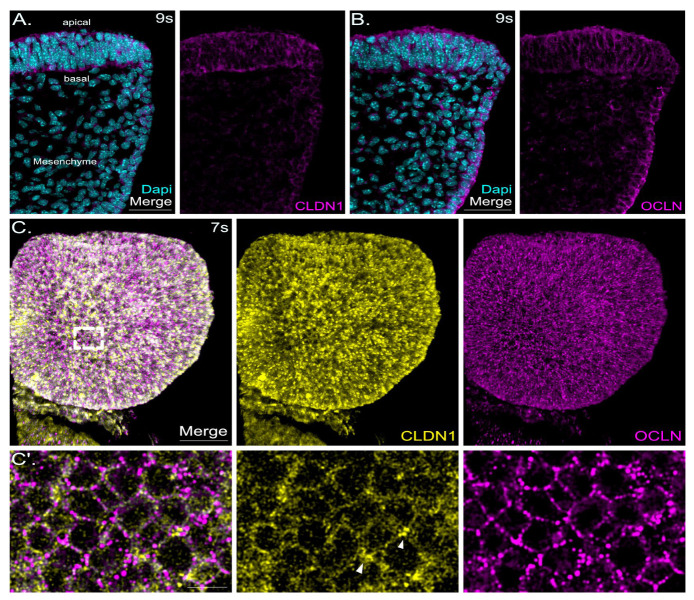
Claudin-1 and occludin are localized to the neuronal and non-neuronal ectoderm in the apical junctions. (**A**): Confocal images show a representative detail of one hemisphere of a coronal section of mouse cranial neural folds with immunofluorescence staining detecting claudin-1 (CLDN1, magenta) at embryonic stage E 8.5 (9 somites, 9s). Nuclei are stained with DAPI (cyan). Merged- and CLDN1 single-channel images are shown at the left and right, respectively. Scale bar: 50 µm. Claudin-1 showed clear apical junction signals and also basolateral localization in the neuroectoderm; less intense signals were detected in the non-neuronal ectoderm and mesenchymal cells. (**B**): Confocal images show a representative detail of one hemisphere of a coronal section of mouse cranial neural folds with immunofluorescence staining detecting occludin (OCLN, magenta) at embryonic stage E 8.5 (9 somites, 9s). Nuclei are stained with DAPI (cyan). Merged- and OCLN single-channel images are shown at the left and right, respectively. Scale bar: 50 µm. Occludin showed clear signals for apical junction localization and less pronounced signals for basolateral localization in the neuroectoderm and in the non-neuronal ectoderm. Faint signals were detected in mesenchymal cells. (**C**): E 8.0 whole-mount mouse embryos (7 somites, 7s) were immunofluorescence co-labelled for claudin-1 (yellow) and occludin (OCLN, magenta). The overlay image of both channels (merge) is shown at the left of panel (**C**), followed to the right by the individual channel images showing the signals for CLDN1 (yellow) and OCLN (magenta). The frontal view on the left hemisphere of forebrains, imaged using confocal microscopy, is shown. Scale bar: 50 µm. (**C**′): Higher-magnification images of boxed area indicated in (**C**) above. Scale bar: 10 µm. En-face view of the neuroepithelium showed a more punctate pattern for occludin signals compared to a more mesh-like pattern displayed by signals for claudin-1. Claudin-1-specific signals were most intense at the multicellular junctions (arrow heads in CLDN1 single-channel image in panel (**C**′)).

**Figure 7 ijms-25-01426-f007:**
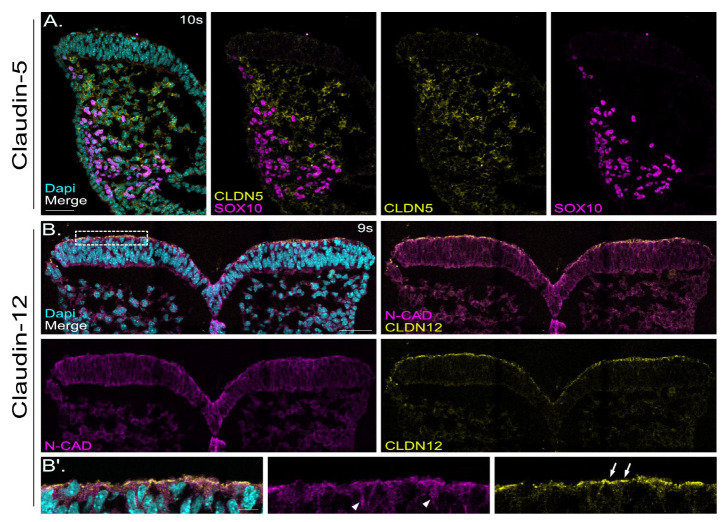
Localization of claudin-5 and claudin-12 in murine cranial neural folds at E 8.5. (**A**): Confocal images show a representative detail of one hemisphere of a coronal section of mouse cranial neural folds with immunofluorescence co-staining detecting the neural crest marker SOX10 (magenta) and claudin-5 (CLDN5, yellow) at embryonic stage E 8.5 (10 somites: 10s). Nuclei are stained with DAPI (cyan). The overlay image of all channels (merge) is shown at the left of panel (**A**), followed to the right by the overlay image for CLDN5 and SOX10 and the individual channel images, showing the signals for CLDN5 (yellow) and SOX10 (magenta). Scale bar: 50 µm. Claudin-5 showed localization in the SOX10-negative mesenchymal cells, underlying the neuroepithelium (neuronal ectoderm), and only a few signals in SOX10-positive neural crest cells (labelled in magenta). (**B**): Confocal images show a representative coronal section of mouse cranial neural folds with immunofluorescence co-staining detecting N-cadherin (N-CAD, magenta) and claudin-12 (CLDN12, yellow) at embryonic stage E 8.5 (9 somites: 9s). Nuclei are stained with DAPI (cyan). The overlay image of all channels (merge) is shown at the top left of panel (**A**), followed to the right by the overlay image for N-CAD and CLDN12. The individual channel images, showing the signals for N-CAD (magenta) and CLDN12 (yellow), are shown below. Scale bar: 50 µm. Claudin-12 showed exclusive localization in the neuroectoderm. (**B**′): Higher magnification of boxed area in B. The overlay image of all channels (merge) is shown at the left of panel (**B**′), followed to the right by the individual channel images, showing the signals for N-CAD (magenta) and CLDN12 (yellow). Claudin-12 displayed clear apical junction localization (arrows in CLDN12 single-channel image), whereas N-cadherin was also found in basolateral domains (arrow heads in N-CAD single-channel image). Scale bar: 10 µm.

**Table 1 ijms-25-01426-t001:** List of antibodies.

Antibody	Manufacturer, Catalog No.	Dilution
Mouse Claudin-1	Santa Cruz Biotechnology, Heidelberg, Germany, sc166338	1:100
Mouse E-cadherin	Abcam, Cambridge, UK, ab76055	1:100
Mouse Lamin B1	Proteintech, Manchester, UK, 66095-1-Ig	1:200
Mouse ZO-1	Invitrogen GmbH, Karlsruhe, Germany, 33-9100	1:100
Rabbit Claudin-3	Invitrogen, 34-1700	1:100
Rabbit Claudin-4	Invitrogen, 36-4800	1:100
Rabbit Claudin-5	Invitrogen, 1548773A	1:100
Rabbit Claudin-12	IBL International, Hamburg, Germany, 18801	1:100
Rabbit ITGA4	Cusabio, Houston TX, USA, CSB-PA011867LA01HU	1:200
Rabbit Occludin	Proteintech, 13409-1-AP	1:200
Goat N-cadherin	Santa Cruz, sc31030	1:100
Goat SOX10	R&D Systems, Wiesbaden, Germany, AF2864	1:200
Donkey anti-mouse Alexa Fluor 488	Abcam, ab150109	1:500
Donkey anti-rabbit Alexa Fluor 555	Abcam, ab150074	1:500
Donkey anti-goat Alexa Fluor 647	Abcam, ab150131	1:500

## Data Availability

Raw RNA sequencing datasets have been deposited in the European Nucleotide Archive (ENA) at EMBL-EBI under accession number PRJEB46911.

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
