# Peer review of "Canonical and Non-Canonical Localization of Tight Junction Proteins during Early Murine Cranial Development"

_ijms, 2024, doi:10.3390/ijms25031426_

Round 1

Reviewer 1 Report

Comments and Suggestions for Authors

In this manuscript, the authors characterized tight junction protein expression in early phases of the developing brain, using a previously generated RNAseq analysis and immunofluorescence microscopy. The analysis revealed different patterns of claudin, ZO-1 and occludin expression, including an unexpected localization of claudin-3 to nuclei. The data represent the first analysis in the mouse and are likely to help drive future investigation of roles for these proteins in early brain development.

Specific Comments

1. The white boxes indicating areas in figures that are magnified were difficult to see, especially in Figure 4. If the box lines were thicker and brighter, that would help.

2. Supplemental Figure 1 could be incorporated into the main body of the text

3. Given the high levels of cldn6 mRNA expression, it was surprising that claudin-6 was not analyzed by immunofluorescence. Including images showing claudin-6 localization would strengthen the study. If this is not practical, it should be acknowledged that claudin-6 expression is high and possible roles for cldn6 in brain development could be discussed in the context of current literature.

4. The RNAseq data is only briefly presented in the results section. A little more detail would be helpful.  Also, there needs to be a call-out to Figure 1 in this section. I was initially surprised that cldn11 was low. It's not critical, but it could be mentioned that cldn11 is not detected because oligodendrocytes are not present at E9.5 and do not appear until E12.5

5. Jam3 and Tjal1 are also highly expressed at the RNA level and not discussed. It would be worth noting their presence. This could be done in the context of the following two papers:

https://pubmed.ncbi.nlm.nih.gov/34880207/
https://pubmed.ncbi.nlm.nih.gov/30617994/

Which mention that miR-212/132 targets cldn1, Jam3 and Tjap1 and impairs blood brain barrier function.

6. Lines 352-354: "There is increasing evidence that claudin proteins function beyond cell-cell adhesion and barrier formation [23]." The Hagen paper (ref 23) has a fairly detailed analysis of putative NLS sequences in claudins. Some discussion of this and specific roles for nuclear localized claudins should be added to the manuscript.

7. Other examples of nuclear localization of cldns that could be cited include:

https://pubmed.ncbi.nlm.nih.gov/33386991/
https://pubmed.ncbi.nlm.nih.gov/29285188/
https://pubmed.ncbi.nlm.nih.gov/15965503/
https://pubmed.ncbi.nlm.nih.gov/9346908/

8. Line 311: replace "Caudins" with "Claudins"

9. Lines 328-330: "Of note, claudin-4 has been associated with the Williams-Beuren syndrome, a neurodevelopmental disorder, with unusual facial features including a broad forehead, underdeveloped chin, and short nose [29]. This needs more explanation. In that paper, they describe that a chromosomal deletion including cldn3 and cldn4 is associated with WBS. It should also be made clear that WBS is not associated with cldn4 mutations.

10. Lines 358-360: "Homozygous mutation of the claudin-5 gene in mice results in size-selective loosening of the blood-brain barrier." Please site https://pubmed.ncbi.nlm.nih.gov/12743111/ here.

11. The Discussion abruptly ends on line 416.  Please add a summary paragraph tying together the observations and future directions of the study.

12. Lines 128, 130:  "Figure B" and "Figure B'" should be "Figure 3 B" "Figure 3 B'"

Comments on the Quality of English Language

Generally good, only a few minor typos and grammatical errors

Author Response

Response to the reviewers:

Reviewer # 1

In this manuscript, the authors characterized tight junction protein expression in early phases of the developing brain, using a previously generated RNAseq analysis and immunofluorescence microscopy. The analysis revealed different patterns of claudin, ZO-1 and occludin expression, including an unexpected localization of claudin-3 to nuclei. The data represent the first analysis in the mouse and are likely to help drive future investigation of roles for these proteins in early brain development.

  1. A.H.: We express our gratitude to the reviewer for providing constructive and valuable comments and suggestions. The manuscript has undergone thorough revisions in response to the received feedback. Alterations have been highlighted using red font color for easy identification. A detailed point-by-point response is presented below.

Specific Comments

1. The white boxes indicating areas in figures that are magnified were difficult to see, especially in Figure 4. If the box lines were thicker and brighter, that would help.

A. H.: We made the lines of the boxes thicker in all Figures.

2. Supplemental Figure 1 could be incorporated into the main body of the text

A.H.: The revised manuscript has the previous Supplementary Figure now included in the main body of the text as Figure 5.

3. Given the high levels of cldn6 mRNA expression, it was surprising that claudin-6 was not analyzed by immunofluorescence. Including images showing claudin-6 localization would strengthen the study. If this is not practical, it should be acknowledged that claudin-6 expression is high and possible roles for cldn6 in brain development could be discussed in the context of current literature.

A.H.: Unfortunately, for claudin-6, the available antibody did not work in our immunohistochemistry approach and therefore further data on claudin-6 localization in the developing cranial neural tube are currently not available. We will certainly follow up on claudin-6 as a potentially important player in the developing forebrain. We included a discussion on the potential role of claudin-6 in the neuroepithelium based on the literature under section 3.1. Data from the literature suggest that claudin-6 could play a role in maintaining pluripotent character of cells and proliferation.

4. The RNAseq data is only briefly presented in the results section. A little more detail would be helpful.  Also, there needs to be a call-out to Figure 1 in this section.

A.H.: Thank you for this suggestion. We have now extended the description on the RNA sequencing data in the result part referring to Figure 1.

I was initially surprised that cldn11 was low. It's not critical, but it could be mentioned that cldn11 is not detected because oligodendrocytes are not present at E9.5 and do not appear until E12.5

A.H.: Thank you for this information. We have now added information on the oligodendrocyte-specific expression of claudin-11 in the discussion under section 3.1.

5. Jam3 and Tjal1 are also highly expressed at the RNA level and not discussed. It would be worth noting their presence. This could be done in the context of the following two papers:
https://pubmed.ncbi.nlm.nih.gov/34880207/
https://pubmed.ncbi.nlm.nih.gov/30617994/
Which mention that miR-212/132 targets cldn1, Jam3 and Tjap1 and impairs blood brain barrier function.

A.H.: We included the literature mentioned by the reviewer and highlighted the role of JAM3 and TJAP1 for the blood-brain barrier in the discussion under section 3.1.

6. Lines 352-354: "There is increasing evidence that claudin proteins function beyond cell-cell adhesion and barrier formation [23]." The Hagen paper (ref 23) has a fairly detailed analysis of putative NLS sequences in claudins. Some discussion of this and specific roles for nuclear localized claudins should be added to the manuscript.

7. Other examples of nuclear localization of cldns that could be cited include:

https://pubmed.ncbi.nlm.nih.gov/33386991/
https://pubmed.ncbi.nlm.nih.gov/29285188/
https://pubmed.ncbi.nlm.nih.gov/15965503/
https://pubmed.ncbi.nlm.nih.gov/9346908/

A.H.: Regarding points 6 and 7, we extended the discussion on potential roles of nuclear localization of claudin-3 and included literature on other examples of nuclear localization of claudins, including claudin-1 in section 3.4.2 of the discussion.

8. Line 311: replace "Caudins" with "Claudins"

A.H.: Thank you for pointing out this spelling mistake. We corrected it.

9. Lines 328-330: "Of note, claudin-4 has been associated with the Williams-Beuren syndrome, a neurodevelopmental disorder, with unusual facial features including a broad forehead, underdeveloped chin, and short nose [29]. This needs more explanation. In that paper, they describe that a chromosomal deletion including cldn3 and cldn4 is associated with WBS. It should also be made clear that WBS is not associated with cldn4 mutations.

A.H.: Thank you for clarifying this. We apologize for this unclear statement on claudin-4 / claudin-3 and its association with WBS and corrected it accordingly also by adding more information (section 3.4.1).

10. Lines 358-360: "Homozygous mutation of the claudin-5 gene in mice results in size-selective loosening of the blood-brain barrier." Please site https://pubmed.ncbi.nlm.nih.gov/12743111/ here.

A.H.: We added the mentioned reference (section 3.4.3, reference 108).

11. The Discussion abruptly ends on line 416.  Please add a summary paragraph tying together the observations and future directions of the study.

A.H.: We now added a paragraph at the end of the discussion summarizing the results and providing a short outlook on future directions.

12. Lines 128, 130:  "Figure B" and "Figure B'" should be "Figure 3 B" "Figure 3 B'"

A. H.: Sorry for this mistake. It is corrected now.

Reviewer 2 Report

Comments and Suggestions for Authors

In the manuscript “Canonical and Non-Canonical Localization of Tight Junction Proteins During Early Murine Cranial Development” Authors found that the diverse tissue and subcellular distribution of tight junction proteins, emphasizing the need for precise regulation during the dynamic processes of forebrain development. The paper is well composed and can be recommended to be published in this journal after minor revisions.

1.       The sentences can be short for better understanding of the readers. For example, in the abstract, “During neurulation the neuronal and non-neuronal ectoderm undergo significant changes and precise coordination of both, epithelial integrity and epithelial dynamics, is essential for accurate tissue morphogenesis” can be written as During neurulation the neuronal and non-neuronal ectoderm undergo significant changes. The precise coordination of both epithelial integrity and epithelial dynamics is essential for accurate tissue morphogenesis. Please check it thoroughly.

2.       History of tight junction proteins and their role. Include more references in the introduction.

3.       A, B, C, D, and E is not clear in Figure 2 and so on. Make it clear.

4.       Conclusion section is missing after discussion.

Author Response

Response to the reviewers:

Reviewer # 2

In the manuscript “Canonical and Non-Canonical Localization of Tight Junction Proteins During Early Murine Cranial Development” Authors found that the diverse tissue and subcellular distribution of tight junction proteins, emphasizing the need for precise regulation during the dynamic processes of forebrain development. The paper is well composed and can be recommended to be published in this journal after minor revisions.

A.H.: We express our gratitude to the reviewer for providing constructive and valuable comments and suggestions. The manuscript has undergone thorough revisions in response to the received feedback. Alterations have been highlighted using red font color for easy identification. A detailed point-by-point response is presented below.

1. The sentences can be short for better understanding of the readers. For example, in the abstract, “During neurulation the neuronal and non-neuronal ectoderm undergo significant changes and precise coordination of both, epithelial integrity and epithelial dynamics, is essential for accurate tissue morphogenesis” can be written as During neurulation the neuronal and non-neuronal ectoderm undergo significant changes. The precise coordination of both epithelial integrity and epithelial dynamics is essential for accurate tissue morphogenesis. Please check it thoroughly.

A.H.: Thank you for the suggestion. We have edited the abstract accordingly and tried to shorten sentences throughout the entire text.

2. History of tight junction proteins and their role. Include more references in the introduction.

A.H.: We have extended the introduction considerably and included multiple references that highlight the history and function of tight junction complex proteins in health and disease.

3. A, B, C, D, and E is not clear in Figure 2 and so on. Make it clear.

A.H.: We have expanded the figure legends to provide more detailed explanations for the images presented in panels A, B, C, etc., for all figures. Specifically, for all immunofluorescence images, the description of images displaying overlay (merged) channels and single channels was previously omitted and has now been included. Additionally, we reverted the formatting of the figure legends to our original layout, which was altered in the IJMS version, and increased the font size for the letters. This, we believe, enhances the clarity of the panels in the figures.

4. Conclusion section is missing after discussion.

A.H.: A concluding summary is now added at the end of the discussion.

Round 2

Reviewer 1 Report

Comments and Suggestions for Authors

The authors did a nice job responding to the critiques, I have no additional concerns